# Cost-effectiveness and cost-utility of a digital technology-driven hierarchical healthcare screening pattern in China

Xiaohang Wu[1,9], Yuxuan Wu[1,9], Zhenjun Tu[2,9], Zizheng Cao[1], Miaohong Xu[1], Yifan Xiang[1], Duoru Lin[1], Ling Jin[1], Lanqin Zhao[1], Yingzhe Zhang[3], Yu Liu[4], Pisong Yan[1], Weiling Hu[1], Jiali Liu[1], Lixue Liu[1], Xun Wang[1], Ruixin Wang[1], Jieying Chen[2], Wei Xiao[1], Yuanjun Shang[1], Peichen Xie[1], Dongni Wang[1], Xulin Zhang[1], Meimei Dongye[1], Chenxinqi Wang[1], Daniel Shu Wei Ting[5,6], Yizhi Liu[1,10] ✉, Rong Pan[2,10] ✉ & Haotian Lin[1,7,8,10] ✉

Utilization of digital technologies for cataract screening in primary care is a potential solution for addressing the dilemma between the growing aging population and unequally distributed resources. Here, we propose a digital technology-driven hierarchical screening (DH screening) pattern implemented in China to promote the equity and accessibility of healthcare. It consists of home-based mobile artificial intelligence (AI) screening, community-based AI diagnosis, and referral to hospitals. We utilize decision-analytic Markov models to evaluate the cost-effectiveness and cost-utility of different cataract screening strategies (no screening, telescreening, AI screening and DH screening). A simulated cohort of 100,000 individuals from age 50 is built through a total of 30 1-year Markov cycles. The primary outcomes are incremental cost-effectiveness ratio and incremental cost-utility ratio. The results show that DH screening dominates no screening, telescreening and AI screening in urban and rural China. Annual DH screening emerges as the most economically effective strategy with 341 (338 to 344) and 1326 (1312 to 1340) years of blindness avoided compared with telescreening, and 37 (35 to 39) and 140 (131 to 148) years compared with AI screening in urban and rural settings, respectively. The findings remain robust across all sensitivity analyses conducted. Here, we report that DH screening is cost-effective in urban and rural China, and the annual screening proves to be the most cost-effective option, providing an economic rationale for policymakers promoting public eye health in low- and middle-income countries.

Digital technologies have brought revolutionary transformations to the healthcare industry, including big data, artificial intelligence (AI), cloud computing, the Internet of Things (IoT), 5th generation (5G) wireless networks, and digital security capabilities such as blockchain[1–5]. The accelerated development of these technologies could be leveraged to improve resource allocations and medical efficiency, especially in low- and middle-income countries (LMICs) where high-quality healthcare resources are scarce or unevenly distributed[6,7]. To fully realize these potential benefits, further innovation of integrated platforms using a combination of technologies remains to be explored.

Previous studies have shown that AI screening, telemedicine screening (telescreening) and AI-telescreening programs are more cost-effective than traditional face-to-face screening mode[8–11]. To further promote the equity and accessibility of healthcare, a digital technology-driven hierarchical (DH) screening pattern has been proposed. It consists of AI screening and diagnosis models based on multiple examination devices, with cloud computing and IoT facilitating telecommunications[12–14]. The Zhongshan Ophthalmic Center (ZOC), a prestigious ophthalmic hospital in China, has implemented this approach since 2018. DH screening comprises three steps for common ophthalmic disease management. In the first step, participants are instructed to take ocular pictures at home via mobile terminals for AI screening. Second, individuals showing signs of potential eye conditions are directed to community facilities where they undergo comprehensive AI-based diagnostic assessments. Finally, patients identified as requiring further medical attention based on positive AI results are referred to specialized tertiary hospitals for appropriate treatments (Fig. 1). The DH screening pattern has substantially promoted medical service capability compared with traditional face-to-face mode, with the potential to be further applied to screening, monitoring, and managing eye diseases in a home-based setting[6]. However, the economic effects remain to be evaluated to support policymakers' decisions regarding promoting the application at scale.

Cataracts are the leading cause of visual impairment worldwide, accounting for more than half of blindness in LMICs[15]. With the global trend of population aging, the number of cataract blindness cases in China is projected to reach 20 million by 2050[16]. Early diagnosis and timely management of cataracts are essential for improving patients' quality of life and reducing healthcare burdens[17]. However, the medical resource distributions are far from satisfactory, particularly in primary healthcare within LMICs[18]. The DH screening pattern achieves high accuracy and provides real-time referable advice for patients with cataracts[6]. Additionally, moderate to severe cataracts are visible through external appearances captured by mobile terminals, making it possible for home-setting screening on a large scale[19]. Therefore, cataracts were chosen as a case study to conduct an economic analysis of the DH screening, hoping to fill the evidence gap and promote its application.

In this study, we build decision-analytic Markov models to analyze the cost-utility and cost-effectiveness of DH screening and compared it with no screening, telescreening, and AI screening with different frequencies in urban and rural China (Supplementary Fig. 1). The primary outcomes are incremental cost-utility ratios (ICURs) and incremental cost-effectiveness ratios (ICERs). One-way and probabilistic sensitivity analyses are performed to test the robustness of the results. The study shows that DH screening dominates no screening, telescreening and AI screening in urban and rural China. Annual DH screening emerges as the most economically effective strategy compared with telescreening and AI screening. The findings remain robust across all sensitivity analyses conducted. The following results show that DH screening proves to be a cost-effective strategy in urban and rural China, which serves as a practical reference for policymakers and healthcare service providers in LMICs.

## Results
### Different screening strategies vs. no screening
The mean expected medical costs for a participant in the next 30 years were $2236 (2227 to 2244), the QALYs gained were 14.31732 (14.31271 to 14.32192), and the expected years of blindness were 0.40993 (0.40924 to 0.41063) in the urban setting. The values for rural areas were $2913 (2902 to 2924), 13.65998 (13.65572 to 13.66423), and 0.56266 (0.56204 to 0.56327), respectively. The cost-effectiveness and cost-utility analysis showed that telescreening, AI screening and DH screening were all dominant over no screening in urban and rural settings (Table 1 and Fig. 2).

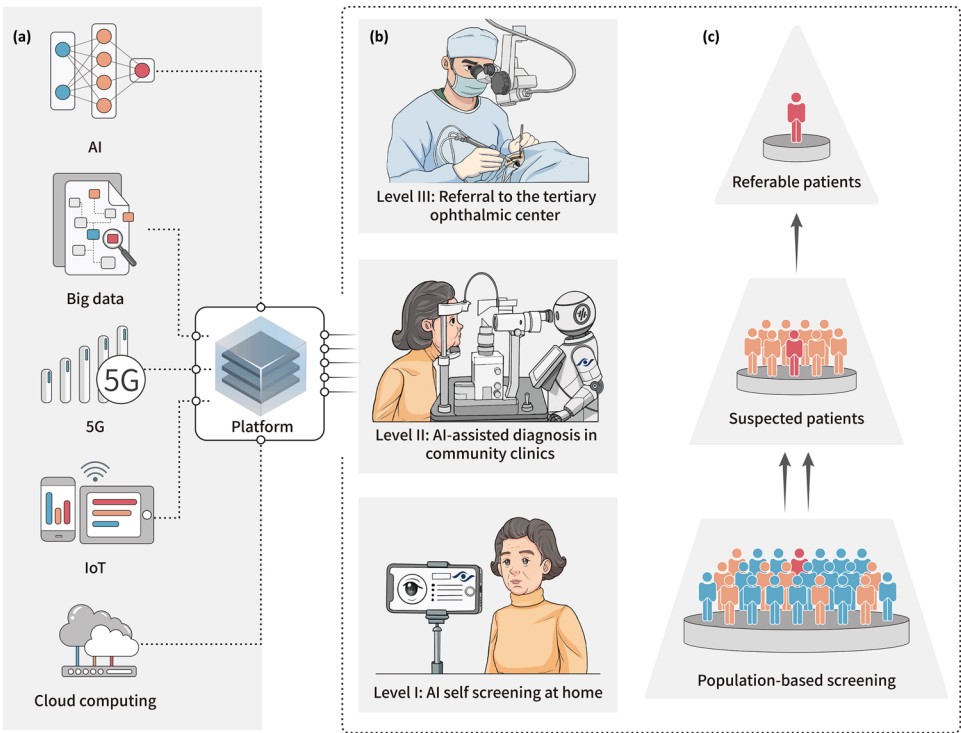

**Fig. 1 | Diagram of the digital hierarchical (DH) screening pattern.**
**a** Compositions of the digital technologies in the hierarchical screening platform.
**b** The workflow of DH screening and referral patterns. **c** AI screening and diagnosis through the IoT and community-based clinics help distinguish suspected patients and suggest referable patients to tertiary eye centers for comprehensive treatment. DH screening digital hierarchical screening. AI artificial intelligence, 5G 5th generation wireless networks, IoT internet of things.

**Table 1 | Base-case cost-effectiveness and cost-utility results of different screening strategies in urban and rural settings**

| | Comparison screening interval for ICER calculation | Costs per person, $ | QALYs per person | Incremental costs per 100,000 people screened, $ | Cost-utility Incremental QALYs per 100,000 people screened | ICURs (95% CI), $ | Years of blindness per person | Cost-effectiveness Years of blindness avoided per 100,000 people screened | ICERs (95% CI), $ |
|---|---|---|---|---|---|---|---|---|---|
| **Urban** | | | | | | | | | |
| No screening | \ | 2236 | 14.31732 | \ | \ | \ | 0.40993 | \ | \ |
| Telescreening | No screening | 2215 | 14.33164 | −2,099,312 | 1433 | Dominating | 0.39921 | 1073 | Dominating |
| AI screening | No screening | 2197 | 14.33589 | −3,806,182 | 1858 | Dominating | 0.39605 | 1389 | Dominating |
| DH screening | No screening | 2189 | 14.33654 | −4,628,416 | 1923 | Dominating | 0.39556 | 1438 | Dominating |
| | Telescreening | \ | \ | −2,529,104 | 490 | Dominating | \ | 365 | Dominating |
| | AI screening | \ | \ | −822,234 | 65 | Dominating | \ | 49 | Dominating |
| **Rural** | | | | | | | | | |
| No screening | \ | 2913 | 13.65998 | \ | \ | \ | 0.56266 | \ | \ |
| Telescreening | No screening | 2806 | 13.70500 | −10,708,193 | 4503 | Dominating | 0.52060 | 4205 | Dominating |
| AI screening | No screening | 2750 | 13.71655 | −16,372,083 | 5658 | Dominating | 0.50991 | 5275 | Dominating |
| DH screening | No screening | 2737 | 13.71855 | −17,608,255 | 5857 | Dominating | 0.50804 | 5462 | Dominating |
| | Telescreening | \ | \ | −6,900,061 | 1354 | Dominating | \ | 1256 | Dominating |
| | AI screening | \ | \ | −1,236,171 | 199 | Dominating | \ | 187 | Dominating |

Costs are expressed in US dollars. Costs, QALYs, and years of blindness are defined as lifetime values per person. Incremental costs, incremental QALYs, ICURs, years of blindness avoided, and ICERs are defined as values per 100,000 people. The ICER thresholds of cost-effectiveness are $31,656 and $41,757 per QALY gained for rural and urban settings, respectively. The ICER thresholds of being highly cost-effective are $10,552 and $13,919 per QALY gained for rural and urban settings, respectively. Negative ICUR or ICER is defined as dominating. *ICER* incremental cost-effectiveness ratio. *QALYs* quality-adjusted life-years, *ICUR* incremental cost-utility ratio, *DH screening* digital hierarchical screening.

### One-way and probabilistic sensitivity analysis

To verify the robustness of the results, we conducted an extensive sensitivity analysis, showing that the base-case results were robust to the broad range of parameter values, and the ICURs were consistently less than the per capita GDP of urban and rural areas (Fig. 3). Tornado diagrams showed the parameters that had the greatest influence on the ICURs. In our study, the prevalence and utility of cataracts and the indirect costs of blindness were common parameters in most screening strategies. The results showed that all screening strategies were dominant over no screening, which were robust to randomly distributed parameters in urban and rural settings (Supplementary Fig. 2). By taking 10,000 random draws from the probabilistic sensitivity analysis, the cost-effectiveness acceptability curve diagram showed that DH screening was the best strategy, accounting for 76.64% and 74.77% of the simulations in urban and rural areas under the current WTP thresholds (Fig. 4).

### DH screening vs. telescreening and AI screening with different frequencies

We evaluated the cost-effectiveness and cost-utility of DH screening compared with telescreening and AI screening. The expected medical costs for a participant in DH screening were $2189 (2181 to 2197), the QALYs gained were 14.33654 (14.33191 to 14.34117), and the expected years of blindness were 0.39556 (0.39487 to 0.39625) in the urban setting. The data for rural settings were $2737 (2727 to 2747), 13.71855 (13.71417 to 13.72292), and 0.50804 (0.50739 to 0.50869), respectively. The DH screening dominated telescreening and AI screening (Table 1). We compared different frequencies to derive the most cost-effective screening strategy (Fig. 2). For a screening range from once-off, every 5 years to every year, a series of comparisons with the former interval showed that annual screening was the most cost-effective strategy in DH screening (Table 2). Furthermore, annual DH screening proved to be the most cost-effective strategy compared to telescreening and AI screening (Supplementary Table 1).

### Discussion

In this study, we conducted economic evaluations of DH screening with different screening strategies and found that it was more cost-effective than no screening, telescreening, and AI screening strategies in urban and rural China. Annual screening proved to be the most cost-effective strategy to avoid cataract-related vision loss under the latest WTP thresholds. In broad sensitivity analyses, the main outcomes remained robust to a wide range of changes in parameters.

The traditional face-to-face screening and referral mode remains mainstream in LMICs, requiring enormous amounts of manpower and specialized equipment, thus compromising medical coverage and efficiency. With further consolidation of telehealth in recent years, telescreening has been widely accepted and has facilitated large-scale screening in remote areas where high-quality medical services are lacking. Recent studies have shown the cost-effectiveness of telescreening in common eye disease screening[10,20,21]. The rapid development of AI technologies has brought novel breakthroughs in medical industries, significantly improving healthcare efficiency and resource utilization. Evidence indicates that AI screening is cost-effective in screening retinopathy of prematurity, diabetic retinopathy (DR), melanoma, dental caries, precancerous polyps, etc.[22,23]. Xie and colleagues conducted a cost-minimization analysis to prove the superiority of semi-automated AI telescreening of DR[9]. Recently, Liu et al. proved that AI telescreening was cost-effective in multiple eye disease screening in China[11]. Currently, the development of mobile health (mHealth) and the IoT facilitates screening is undergoing a rapid transition toward decentralization, where some or all health assessments are performed remotely in participants' homes instead of in medical centers[24]. Numerous diseases have the potential to be screened by deep learning using photography or video on smartphones as a diagnostic tool, including melanoma, scoliosis, certain ocular disorders and related systematic diseases[25–27]. A previous study showed the potential of cataract screening through anterior segment pictures captured by smartphones[13]. A deep learning screening model for infant vision impairment has recently been developed, allowing parents to detect children's vision disorders by recording their gazing behaviors through smartphones at home with high accuracy[13]. In our study, a DH screening pattern was implemented further promoting the accessibility of vision screening in China. Our results showed that the pattern could obtain even better economic returns than telescreening

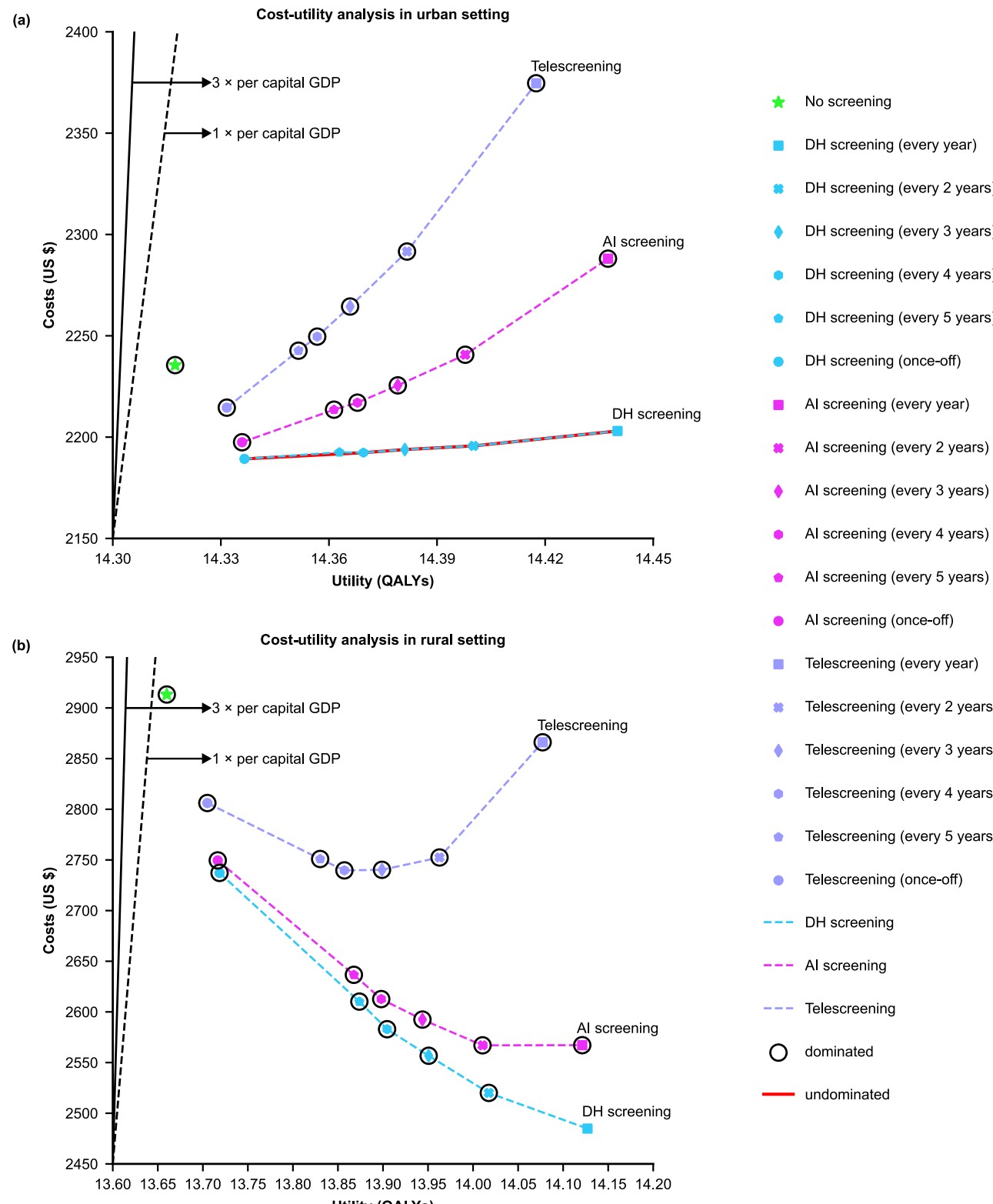

**Fig. 2 | Cost-utility curves for different screening strategies in China. a** Cost-utility analysis in urban setting. **b** Cost-utility analysis in rural setting. Black solid line = Three times the per capita GDP for the highly cost-effective frontier. Black dashed line = One times the per capita GDP for the cost-effective frontier. Green dot = no screening. Purple dots and dashed line = telescreening with different frequencies. Pink dots and dashed line = AI screening with different frequencies. Blue dots and dashed line = DH screening with different frequencies. Strategies on the cost-effective frontier dominate strategies above the frontier. GDP gross domestic product, AI artificial intelligence, QALYs quality-adjusted life-years, DH screening digital hierarchical screening. Source data are provided as a Source Data file.

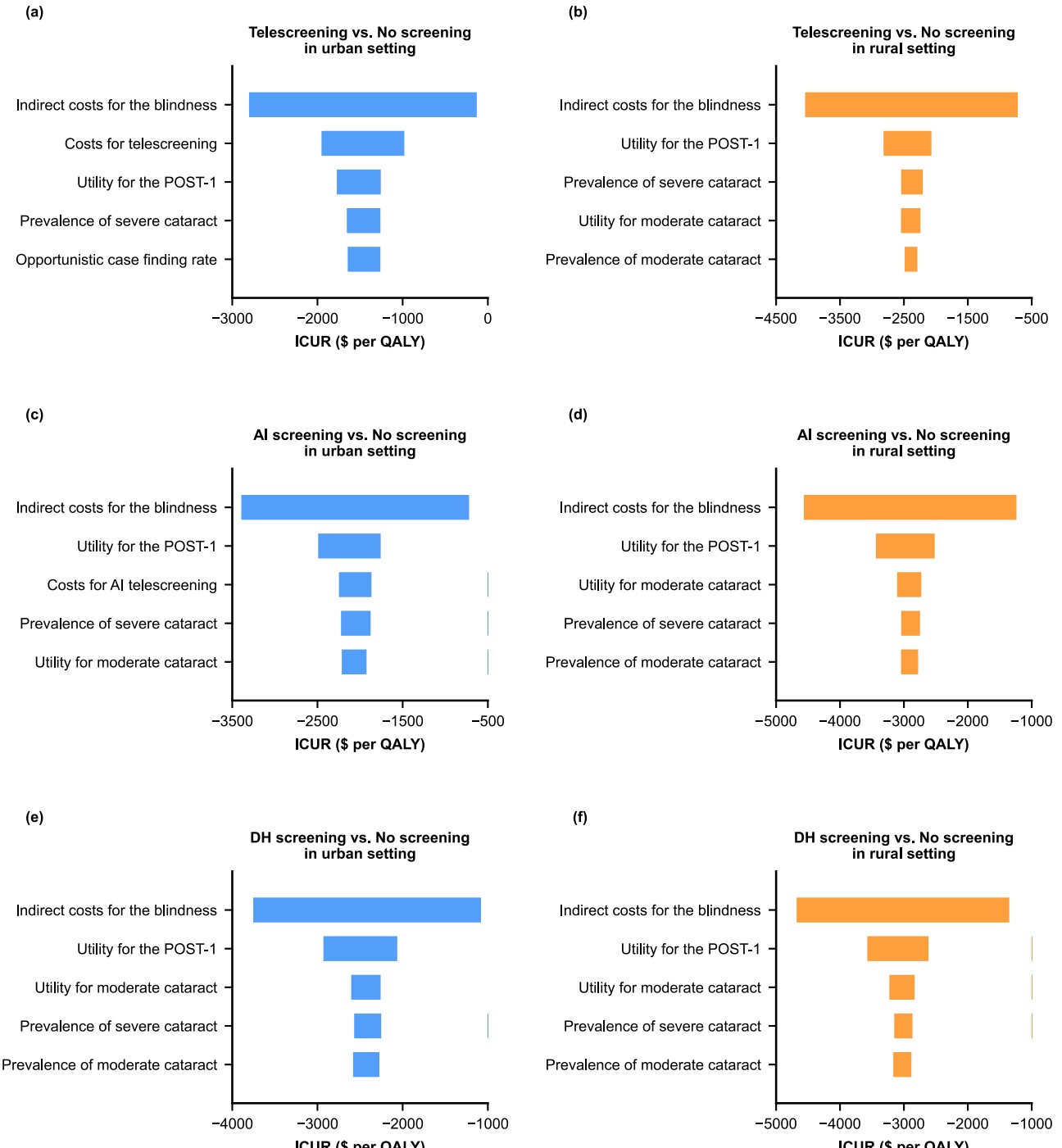

**Fig. 3 | Deterministic one-way sensitivity analysis.** Costs are expressed in US dollars. The top five parameters that caused the greatest impact on the ICURs are shown in the above figures. We performed one-way sensitivity analyses for telescreening vs. no screening (**a** and **b**), AI screening vs. no screening (**c** and **d**), and DH screening vs. no screening (**e** and **f**) in urban (**a**, **c**, **e**) and rural (**b**, **d**, **f**) settings. The thresholds of cost-effectiveness were $31,656 and $41,757 per QALY gained for rural and urban settings, respectively. The intervention was dominant if the ICUR value was negative. DH screening digital hierarchical screening, GDP gross domestic product, ICUR incremental cost-utility ratio, QALYs quality-adjusted life-years, AI artificial intelligence. Source data are provided as a Source Data file.

and AI screening. The superiority is mainly attributed to easy access to smartphone AI screening, increased referral compliance, and the substantial reduction in human assessment time and labor costs without sacrificing accuracy due to further community-based AI confirmation. The DH screening system has brought primary screening into home settings through mobile terminals, which are much more accessible to the population, especially in remote areas. Additionally, participation through the IoT and AI can help address patient retention challenges and promote the compliance of residents who need regular

screening and further referral[7]. After AI self-screening at home, only suspected patients were referred for further examinations, thus effectively avoiding costs related to unnecessary referrals and examinations.

Previous economic evaluations of eye screening mainly focused on glaucoma, DR and age-related macular degeneration (AMD) which cause irreversible vision impairment and blindness[9,10,28]. Most strategies were reported to be cost-effective in population-based screening. However, in this study, we find that all cataract screening strategies dominate no

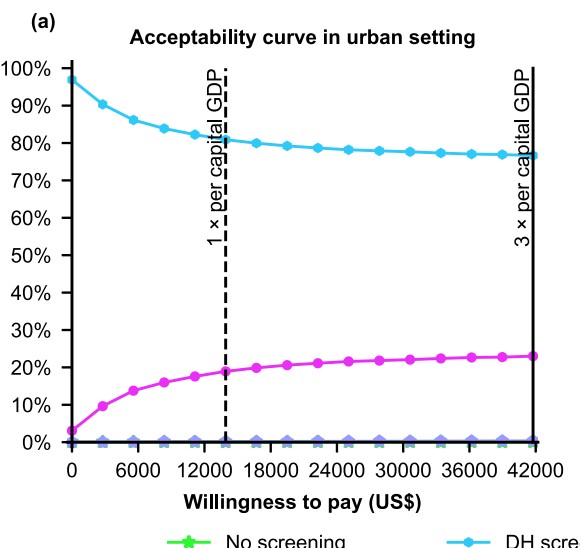

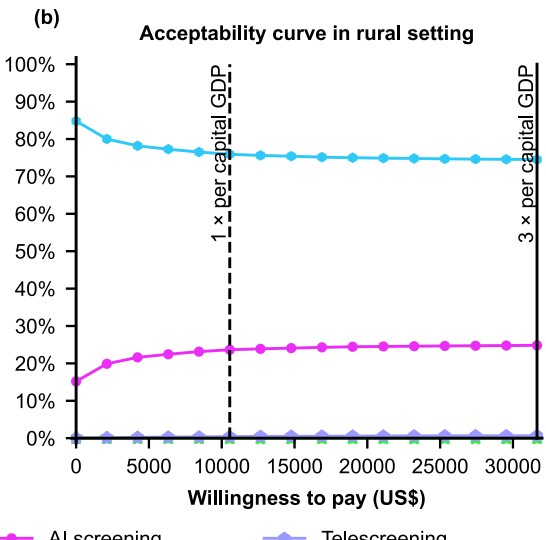

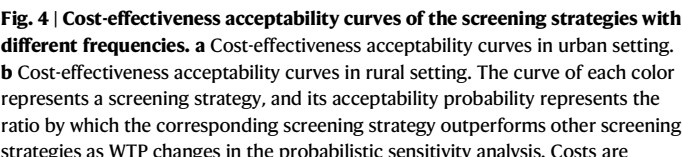

**Fig. 4 | Cost-effectiveness acceptability curves of the screening strategies with different frequencies. a** Cost-effectiveness acceptability curves in urban setting. **b** Cost-effectiveness acceptability curves in rural setting. The curve of each color represents a screening strategy, and its acceptability probability represents the ratio by which the corresponding screening strategy outperforms other screening strategies as WTP changes in the probabilistic sensitivity analysis. Costs are expressed in US dollars. Solid line = Three times the per capita GDP for the highly cost-effective frontier. Dashed line = One times the per capita GDP for the cost-effective frontier. DH screening digital hierarchical screening, GDP gross domestic product, AI artificial intelligence, WTP willingness to pay. Source data are provided as a Source Data file.

screening, achieving greater economic and societal returns. This is attributed to the relatively high prevalence, low screening and treatment costs, and significant postoperative visual improvement of cataracts compared with the foregoing irreversible blindness-causing diseases. Importantly, our findings suggest that annual cataract screening is the most cost-effective option for elderly individuals over 50. From once-off, every 5 years to 1 year, the more frequent the screening strategy is, the more cost-effective it becomes. Aligning with the guidelines indicating that high-risk individuals should have ocular examinations once a year, this annual screening interval is clinically and economically appropriate[29]. Additionally, different trends are shown in urban and rural areas regarding the costs and benefits of DH screening with varying frequencies (Fig. 2). In urban areas, both costs and QALYs increase as the frequency increases since more frequent screening leads to more spending and blindness years avoided. In rural settings, as the screening frequency increases, more benefits are gained while total costs decrease. The main reason is that more frequent DH screening could save more indirect costs due to blindness in rural areas. First, the prevalence of cataracts and the proportions of severe cataracts are higher in rural areas. Second, due to relatively limited medical resources, the opportunistic case finding rate is lower than that in urban settings. Rural patients would benefit from more frequent screening programs for being spotted and treated; hence, total societal spendings were saved by avoiding utility loss and blindness-related indirect costs. The results suggest that DH screening could produce great economic benefits especially in rural China, which can also serve as an example for LMICs with similar settings or epidemiologic characteristics.

This study has several strengths. We propose the pilot DH screening pattern and assess its cost-effectiveness and cost-utility in urban and rural China. Key parameters are derived from real-world investigations and research mostly specific to China and Asian countries. The thresholds are calculated based on the latest data, including the local GDP in 2022, the urbanization rate in 2022, the mortality rate in the China Population Census Yearbook 2020, etc., ensuring the accuracy and effectiveness of the main conclusions. However, there remain limitations to be discussed. First, the process of disease treatment was simplified in illustrating the workflow of the DH screening strategies. Cataracts were selected for analysis as they are recognized

as the leading cause of vision loss and blindness worldwide. Additionally, it was presumed that patients with bilateral cataracts would demonstrate similar levels of severity and postoperative visual outcomes. The costs and utility might vary if dissymmetric cataracts or postoperative complications are considered. Second, comprehensive ocular examinations including the anterior segment and retinal fundus are more practical and efficient in real-world applications. Multiple eye disease screening will be considered in our future research. Although only cataract is considered in the simulated analysis, we would replenish the workflow to address this ethical issue, meanwhile summarize the percentages of missing out those chronic diseases (e.g., glaucoma, AMD) in preparation for the next study involving multiple diseases screening. For instance, once the participants are presented with nonreferral mild cataract but accompanied by unmatched poor vision (suspicious of other ocular diseases), the trained staff in the community-based clinic will give a reminder of further referral and comprehensive examinations, thus avoid missing out other chronic eye diseases. Third, although per capita GDP is the standard way of setting WTP thresholds in cost-effectiveness analysis, the conditions in China are different. China has a smaller proportion of consumption market in GDP than the United States. Thus, the "money in the people's pocket" is not reflected by the GDP in the same way as the other countries and a better way to derive WTP in the Chinese setting is needed.

In conclusion, this study provides initial evidence that the DH screening pattern is more cost-effective than no screening, telescreening and AI screening in China and feasible for further implementation in other fields of medicine worldwide. The results suggest that the combination of digital technologies and mHealth applications could effectively promote public eye health management and quality of life, thus providing a valuable reference for the government and policymakers in LMICs.

## Methods
### Model overview
A decision-analytic Markov model was constructed using TreeAge Pro 2022 (TreeAge Software; Williamstown, MA, USA) for the economic analysis of different screening strategies for cataracts. The model was built on a simulated cohort of 100,000 residents from 50 years

**Table 2 | Cost-effectiveness of different screening intervals in rural and urban settings**

| Screening interval | Comparison screening interval for ICER calculation | Urban | | | | Rural | | | |
|---|---|---|---|---|---|---|---|---|---|
| | | Costs per person, $ | Years of blindness per person | Years of blindness avoided per 100,000 people screened | ICERs, $ | Costs per person, $ | Years of blindness per person | Years of blindness avoided per 100,000 people screened | ICERs, $ |
| **Telescreening** | | | | | | | | | |
| Once-off | \ | 2215 | 0.39921 | \ | \ | 2806 | 0.52060 | \ | \ |
| Every 5 years | Once-off | 2243 | 0.39761 | 160 | 17,598 | 2751 | 0.47131 | 4930 | Dominating |
| Every 4 years | Every 5 years | 2250 | 0.39701 | 60 | 11,497 | 2740 | 0.46076 | 1054 | Dominating |
| Every 3 years | Every 4 years | 2264 | 0.39597 | 104 | 14,277 | 2740 | 0.44544 | 1533 | 37 |
| Every 2 years | Every 3 years | 2292 | 0.39408 | 188 | 14,413 | 2752 | 0.42354 | 2190 | 556 |
| Every year | Every 2 years | 2375 | 0.39017 | 392 | 21,199 | 2866 | 0.39067 | 3286 | 3459 |
| **AI screening** | | | | | | | | | |
| Once-off | \ | 2197 | 0.39605 | \ | \ | 2750 | 0.50991 | \ | \ |
| Every 5 years | Once-off | 2214 | 0.39413 | 192 | 8360 | 2637 | 0.45274 | 5718 | Dominating |
| Every 4 years | Every 5 years | 2217 | 0.39346 | 66 | 5224 | 2613 | 0.44173 | 1101 | Dominating |
| Every 3 years | Every 4 years | 2226 | 0.39237 | 109 | 7798 | 2592 | 0.42640 | 1532 | Dominating |
| Every 2 years | Every 3 years | 2241 | 0.39051 | 186 | 8144 | 2567 | 0.40600 | 2041 | Dominating |
| Every year | Every 2 years | 2288 | 0.38713 | 338 | 14,015 | 2567 | 0.37881 | 2718 | 5 |
| **DH screening** | | | | | | | | | |
| Once-off | \ | 2189 | 0.39556 | \ | \ | 2737 | 0.50804 | \ | \ |
| Every 5 years | Once-off | 2193 | 0.39360 | 196 | 1666 | 2610 | 0.44977 | 5828 | Dominating |
| Every 4 years | Every 5 years | 2192 | 0.39293 | 66 | Dominating | 2583 | 0.43878 | 1099 | Dominating |
| Every 3 years | Every 4 years | 2194 | 0.39184 | 109 | 1405 | 2557 | 0.42360 | 1518 | Dominating |
| Every 2 years | Every 3 years | 2196 | 0.39002 | 183 | 986 | 2520 | 0.40361 | 1999 | Dominating |
| Every year | Every 2 years | 2203 | 0.38676 | 326 | 2262 | 2485 | 0.37742 | 2619 | Dominating |

Costs are expressed in US dollars. Costs and years of blindness are defined as lifetime values per person, whereas years of blindness avoided and ICERs are defined as values per 100,000 people. The ICER thresholds of cost-effectiveness are $31,656 and $41,757 per QALY gained for rural and urban settings, respectively. The ICER thresholds of being highly cost-effective are $10,552 and $13,919 per QALY gained for rural and urban settings, respectively. Negative ICURs and ICERs are defined as dominating. ICER incremental cost-effectiveness rate. AI artificial intelligence. DH screening digital hierarchical screening.

through a total of 30 1-year Markov cycles, which is the common target population based on previous economic evaluations of eye disease screening in elderly individuals[10,11,28]. The participants were allowed to enter the model as either healthy (free from cataracts) or unhealthy (with cataracts) and could progress to death from any health states. The primary outcomes were ICURs and ICERs. We assumed that the severity and postoperative visual acuity for bilateral cataract patients were similar. Based on the clinical practice guidelines, the severity of cataracts is assessed by slit-lamp photographs using LOCS grading standards[30]. Increased cataract severity is strongly associated with a decrease in visual acuity[31]. Therefore, cataract patients' BCVA is one of the common classification methods in clinical trials[11,32,33]. We derived data of mild, moderate and severe cataracts based on patients' best corrected visual acuity (BCVA) > 0.3, 0.1–0.3, and <0.1 respectively from published research[11,34]. Moderate and severe cataracts were identified as referable cataracts[35,36]. A Markov model was constructed to simulate the disease progression of mild and moderate to severe cataracts. During each cycle, the participants either stayed in the same stage or transitioned to the more severe phase (Supplementary Fig. 3). Accordingly, we defined three postoperative groups based on patients' BCVA after surgery, namely the POST-1 group (BCVA > 0.3), the POST-2 group (BCVA 0.1–0.3), and the POST-3 group (BCVA < 0.1) for utility analysis[32,33]. Since there was no significant change in postoperative visual outcomes during the long-year follow-up, we assumed that patients' visual acuity and utility remained stable after surgery[36]. Severe cataracts and the POST-3 group were combined as bilateral blindness for indirect cost calculations[10,28]. We collected data from real-world eye screening programs and a literature search of prevalence, compliance, utility, and other parameters, most of which were specific to China or other LMICs. The costs of screening, examination, and treatment came from real-world eye disease screening programs and the ZOC.

## Screening strategies and scenarios

**No screening.** Cataract patients might be diagnosed and treated upon opportunistically presenting at a hospital for another concern, without routine ophthalmic screening.

**Telescreening.** Residents over 50 were educated and invited to participate in a cataract telescreening in community-based clinics, including the visual acuity test and slit lamp photography. The data were transmitted to the ZOC telemedicine platform. One certificated ophthalmologist assessed the severity and provided an assessment report back to the primary care settings. The participants returned to collect the reports after one week. Once referable cataracts were detected, patients were referred to the ZOC for comprehensive examinations, diagnosis, and treatments. The others were suggested for follow-up.

**AI screening.** Residents over 50 were educated and invited to participate in AI screening in community-based clinics, including the visual acuity test and slit lamp photography. The AI models provided a real-time diagnosis and referable advice. Participants with referable cataracts were referred to the ZOC. The others were suggested for follow-up.

**DH screening.** Residents over 50 were educated and invited to participate in DH screening by using an app for AI cataract screening on smartphones at home. The photographs of ocular anterior segments were captured by themselves or family members as instructed. High-quality images were uploaded for AI diagnosis. Suspected patients were referred to community clinics for visual acuity tests and slit lamp photography assisted by primary eye care staff. Once referable cataracts were detected by AI, patients were referred to the ZOC. The others were suggested for follow-up (Supplementary Fig. 1).

## Cataract prevalence, transition probabilities, and screening performances

The prevalence of senile cataracts is 26.66% and 28.79% in urban and rural areas, respectively, based on the systematic review and meta-analysis of large-scale epidemiological surveys of people over 50 years old in China[37]. The annual transition probabilities were derived from the literature on the natural progression of cataracts in the Chinese population. In studies reporting multiyear incidences, the annual incidence was calculated as $r = -\log(1- p)/t$, where r represents the 1-year incidence and p means the cumulative incidence over interval $t$[38]. (Supplementary Table 2).

The model performances of DH smartphone-based screening and community-based AI/DH screening were derived from an ongoing national cataract AI screening investigation launched by the ZOC in 2018 to promote collaborative efficiency and medical resource coverage[6]. The AI cataract screening model involving multilevel clinical scenarios proved to be robust in a real-world evaluation. In the first stage of smartphone-based screening, the AI model achieved a sensitivity of 88.67% and a specificity of 89.33%. Next, in the community-based screening setting, the AI agent distinguished referable cataracts with a sensitivity of 94.80% and a specificity of 97.00%. The performance of telemedicine screening was collected from previous research and had a sensitivity of 95.00% and a specificity of 97.00%[39]. (Supplementary Table 3).

## Screening and treatment costs

Direct and indirect costs were included in the analysis. Direct costs included ophthalmic screening, examination, treatment, follow-up, transportation, food, and accommodation charges for further visits to specialized hospitals. Indirect costs consisted of one accompanying family member's time and wage loss based on the time spent and per capita daily income in rural and urban areas. The costs of examination, treatment, and follow-up were obtained from the ZOC under the Chinese government's control and varied little from institution to institution. All costs were expressed in US dollars at the exchange rate as of 2 November 2022 (1 USD = 7.2 CNY), listed in Supplementary Table 4.

Screening costs included equipment, labor, and transportation costs. The annualized cost of fixed assets was calculated by assuming a life span of 5 years, collected from the Finance Department and Procurement Center of the ZOC. Since the participants were over 50, we assumed that they did not produce a wage loss (Supplementary Table 5).

Patients with mild cataracts were suggested for follow-up till next screening. For referable patients, cost computation for examination, treatment and follow-up are listed in detail in Supplementary Table 6. For patients with bilateral blindness, the annual economic burden of indirect costs was assumed to be $3600 per person, including loss of labor resources and productivity of caregivers, based on previous research[28].

## Other parameters (compliance, utility, mortality rate, and threshold)

We assumed that 98% of residents had access to a smartphone and could use the app for AI cataract screening on their own or with assistance from family members based on the coverage of mobile phones and 5G network in China[40,41]. Compliance with telescreening and AI screening in community-based clinics was derived from a previous study that indicated 95% compliance in rural and 90% compliance in urban settings[10,11]. Additionally, a randomized controlled trial (RCT) study suggested that the hospital referral adherence of AI screening and traditional screening was 52% and 40%, respectively[7,10,11]. Considering that patients in the DH screening group had received two positive results and referral reminders, once home-based self-screening feedbacks more than other groups, a reasonable higher referral

adherence rate of 62% was used in this group. Compliance of surgical therapy was 91% and 80% in urban and rural settings, respectively[11]. For those who fail to participate in the screening program, or don't adhere to referral or treatment, the possible results can be natural progression of cataracts; otherwise, they can also be diagnosed and treated in opportunistically case finding or next screening cycle[10,42].

The utility of healthy individuals without cataracts was defined as 1[28]. Patients with mild, moderate, and severe cataracts have utility values of 0.60, 0.45, and 0.26, respectively[43]. The utility values of the POST-1, 2 and 3 groups were 0.75, 0.55 and 0.53, respectively, based on previous research[43].

Age-specific mortality was obtained from the China Population Census Yearbook 2020 from the National Bureau of Statistics[44]. According to previous research, increased odds of mortality for patients with cataracts and no difference after surgery were also accounted for (Supplementary Table 3)[45,46]. The discounted cost and utility rate was 3.5% per annum[10,47].

According to the WHO, the definition of being *cost-effective* refers to interventions that cost less than three times the per capita gross domestic product (GDP). The *highly cost-effective* strategy refers to interventions that cost less than the per capita GDP[48]. The per capita GDP was calculated for urban ($13,919) and rural ($10,552) China based on the 2022 overall per capita national GDP ($12,741), urbanization rate (0.65), and urban-rural ratio (2.45) of per capita disposable income using the following formulas[10,11,49]:

The per capita GDP of urban China

$$= \frac{overall\ per\ capita\ national\ GDP}{(1 + \frac{1}{urban\ to\ rural\ ratio\ of\ per\ capita\ disposal\ income}) \times urbanization\ rate},$$

The per capita GDP of rural China

$$= \frac{overall\ per\ capita\ national\ GDP}{(1 + urban\ to\ rural\ ratio\ of\ per\ capita\ disposal\ income)(1 - urbanization\ rate)}.$$

As a result, the thresholds of willingness to pay (WTP) were $41,757 and $31,656 per quality-adjusted life year (QALY) gained for urban and rural China, respectively. Notably, if the ICUR or ICER was negative with fewer costs spent and more benefits gained, the strategy was defined as *dominating*[47].

## Primary outcomes

The primary outcomes were ICURs and ICERs, calculated using the following formulas:

$$ICURs = \frac{incremental\ cost}{QALY\ gained},$$

$$ICERs = \frac{incremental\ cost}{years\ of\ blindness\ avoided}.$$

## Sensitivity analysis

We performed extensive deterministic sensitivity analysis and probabilistic sensitivity analyses to assess the robustness of the main outcomes. Fluctuation ranges of 10% (probability data including prevalence, sensitivity, specificity, utility, transition probability, etc.), 20% (costs of examinations, treatments, follow-up, etc.), and 50% (screening costs and indirect costs for blindness) were set for sensitivity analysis[10]. Tornado diagrams showed the parameters that had the greatest influence on the ICURs. Probabilistic sensitivity analysis evaluated the impact on the results by taking 10,000 random samples from the probability distribution of each parameter. The methods and results conforming to the Consolidated Health Economic Evaluation Reporting Standards were listed in Supplementary Table 7.

## Reporting summary

Further information on research design is available in the Nature Portfolio Reporting Summary linked to this article.

## Data availability

All data used to construct the model are publicly available and referenced. All parameters and their values can be found in Supplementary Tables 2–3 and Supplementary References. Data can only be shared for noncommercial academic purposes and will require a formal data use agreement. Please email all requests for academic use of raw and processed data to corresponding authors at linht5@mail.sysu.edu.cn. For requests from verified academic researchers, access will be evaluated by the data access committee and be granted within one month. Source data are provided with this paper.

## Code availability

The model was constructed using the software TreeAge Pro 2022 (Healthcare version). The model is intended for research purposes and its use is limited to this purpose. The model can be obtained via a request to the corresponding author linht5@mail.sysu.edu.cn. However, access will only be granted if the intended use is limited to noncommercial academic purposes. Access will be evaluated by the code assessment committee and granted within one month. The example of Markov model and the pseudocode for the algorithm analysis used in this paper is provided in the Supplementary Data 1.

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

## Acknowledgements

Funding was provided by the National Natural Science Foundation of China (92368205), National Natural Science Foundation of China (82171035), National Natural Science Foundation of China (12126609), basic scientific research projects of Sun Yat-sen University (23ykcxqt002), the High-level Science and Technology Journals Projects of Guangdong Province (2021B1212010003), Science and Technology Planning Project of Guangdong Province (2023A1111120011), Science and Technology Planning Project of Guangzhou City (2024B03J1233), the Science and Technology Planning Projects of Guangdong Province (2021B1111610006), the Pazhou Lab (Huangpu) Research and Development Project (2023K0601), the Guangdong Provincial Natural Science Foundation for Progressive Young Scholars (2023A1515030170), and the Guangzhou Basic and Applied Basic Research Project (202201011301). The sponsors and funding organizations had no role in the design or performance of this study. The authors greatly appreciate Professor Jiqian Fang and Director Xiang Chen for the statistical guidance.

## Author contributions

Conception and design: X.W., Y.W., Z.T., Y.L., R.P., and H.L. Data collection: Y.W., Z.C., M.X., P.Y., W.H., J.L., and C.W. Analysis and interpretation: X.W., Y.W., Z.T., L.J., L.Z., Y.Z., and Y.L. Figures: Y.W. and Z.T. Manuscript draft and revisions: X.W., Y.X., Z.T., D.L., L.J., L.L., X.W, R.W., J.C., W.X., Y.S., P.X., D.W., X.Z., M.D., D.S.W.T., Y.L., R.P., and H.L. Funding acquisition: X.W. and H.L. X.W. and H.L. have directly accessed and verified the underlying data reported in the manuscript.

## Competing interests

The authors declare no competing interests.

## Additional information

[1]State Key Laboratory of Ophthalmology, Zhongshan Ophthalmic Center, Sun Yat-sen University, Guangdong Provincial Key Laboratory of Ophthalmology and Vision Science, Guangdong Provincial Clinical Research Center for Ocular Diseases, Guangzhou, Guangdong, China. [2]School of Computer Science and Engineering, Sun Yat-sen University, Guangzhou, Guangdong, China. [3]Department of Epidemiology, Harvard T.H. Chan School of Public Health, Boston, Massachusetts, USA. [4]School of Public Health and Management, Guangzhou University of Chinese Medicine, Guangzhou, Guangdong, China. [5]Singapore Eye Research Institute, Singapore National Eye Centre, Singapore, Singapore. [6]Duke-National University of Singapore Medical School, Singapore, Singapore. [7]Hainan Eye Hospital and Key Laboratory of Ophthalmology, Zhongshan Ophthalmic Center, Sun Yat-sen University, Haikou, Hainan, China. [8]Center for Precision Medicine and Department of Genetics and Biomedical Informatics, Zhongshan School of Medicine, Sun Yat-sen University, Guangzhou, Guangdong, China. [9]These authors contributed equally: Xiaohang Wu, Yuxuan Wu, Zhenjun Tu. [10]These authors jointly supervised this work: Yizhi Liu, Rong Pan, Haotian Lin. ✉e-mail: yizhi_liu@aliyun.com; panr@sysu.edu.cn; linht5@mail.sysu.edu.cn

