## [Peer Review File · Nature Communications]

Reviewers' Comments:

Reviewer #1:

Remarks to the Author:

This is an interesting study that evaluated the cost effectiveness of a unique digital hierarchical (DH) screening pattern for cataract in the community. The model involved 100,000 people in the China setting. They concluded that DH screening is cost-effective in urban and rural China, and this provide a solid model for Low- and middle-income countries (LMICs). However, there are several fundamental problems:

1. The DH system was not described with enough details. How would the participants be selected (apart from the age)? Are they participating by volunteering means? If that is the case, would that be any selection bias (e.g. residents that do not think they have any eye problem will not join the programme). What is exactly the rate of detecting mild, moderate, and severe cataract with the DH system. How correlative would that be with the real-world situation.

2. The authors mentioned that the referral adherence of home-based self-screening could be around 62%. However, who about those who did not adhere to the referral? Would there be a delay in treatment that could also be cost if they don't do so? And for those who are screened and are told to have no cataract, would they think they no longer have any eye problem and miss out other eye diseases that could lead to utility loss (e.g. an asymptomatic glaucoma patient who have been told by the DH system that don't have cataract, ended up missing out the diagnosis of glaucoma and the chance of treatment at the early stage, ended up having permanent utility lost because of delay treatment). Did the model consider that? And if it did, what are the percentages of missing out those chronic diseases (e.g. glaucoma, ARMD).

3. There are too many supplementary materials that could confuse the readers and this is probably not the best way to get away from the "table + figure" restriction set by the journal. For instance, the cost eTable 5-8 could have been combined into one table instead of three. Information of eFigure 4 is probably redundant because it commonly known among cost-effectiveness analysis and could be described in the manuscript.

4. The manuscript has a lot of information but the flow of the whole layout and discussion could make it difficult to follow.

5. Though per capita GDP is the standard way of setting the WTP. However, China's GDP make up is quite different from the European countries and the United State of America. Generally, the Europe and US has a larger proportion of the GDP taken up by the consumption market and lesser proportion occupied by investment and infrastructure. The consumption portion is more related to the actual money that the people have and willing to spend. However, this is the other way round in in China; a larger proportion of the GDP is taken up by the investment and infrastructure and lesser proportion is occupied by consumption. Hence, the "money in the people's pocket" is not reflected by the GDP in the same way as the other countries. The standard per capital GDP may not be the best way to derive WTP in the China setting. So much as that the author could derive the WTP of China as per the customary method, the authors should point this out in the discussion.

Other comments

Introduction

(lines 86-87): Although the statement that "mild and severe cataracts are visible through external appearance capture by mobile terminals, making it possible for home-setting screening on a large scale." is supported by previous publication. However, small pupil and poor quality of photo taking (common for untrained hands) are common. Did the authors take those into consideration and the cost that could lead to?

Methods:

(line 209) A Markov model simulated from 50 years for 30 years – Does that fit into the life

expectancy in China? Furthermore, If the 100,000 residents start from 50 years and begin to do the screening, why would a cataract only catch at 80 years old? (Although this is based on previous eye disease screening studies).

How was the severity of cataract assessed exactly? Seems like not merely based on patients BCVA (line 215 to 216)? Because it was also mentioned in the telescreening part that an ophthalmologist assessed the severity of the cataract for feed back to the primary care settings? (lines 233-234) – authors need to clarify about that point.

How about those who were invited but did not participate in DH screening? Percentage of that? If patient deny for DH screening, would that be the cost as well?

Did the author consider the cost of miss out other eye disease (and the utility loss of that) because the participants feel comfortable with the DH system telling them that do not have cataract and did not seek ophthalmologist/optometrist help even if the visual acuity worse?

“We assumed that patients’ visual acuity and utility remained stable after surgery” (line 222) How about PCO after surgery? How about other potential complications. How about complications of cataract and the related management.

Authors should rearrange the flow of describing the study methodology. Information tended to scatter within the section and that could be confusing to the readers.

Given that the cost-effectiveness analysis was based on national cataract AI screening programme implemented, authors should clarify the details and terms related to the programme before describing the cost-effectiveness analysis based on the specific terms.

For instance

- POST-1, POST-2 and POST-3 groups were mentioned under Model Overview (line 219 to 210) with referral to a supplementary figure (eFigure 3), while the details were only described scattered throughout the section. That makes it very difficult to follow.

- They mentioned about “AI performances” (line 254) at the later part of the session. That makes it confusing again because it seems unclear whether they are referring to the AI in the “AI Screening” or the AI in the “DH screening”

The authors did not mention about inflation in the calculation.

Results:

The results part is related to the assumption of the model that the authors made. Therefore, the problems in the results is corresponding to that of the methodology.

Discussion:

The discussion tried to be too inclusive about various aspect of AI screening and divert from what really need to discuss about the results of the study. The authors spent a lot of effort discussing about the advantage of home screening and AI of other eye diseases (without much disadvantage), spending a paragraph to emphasise on the same point, but the discussion of the important parts was a bit anaemic. For instance, (line 177-179) they mentioned that increasing frequency of screening in the rural settings gained more benefits while the cost decrease. The author explained that the disparity (compared with the urban setting) was possible due to

- the difference in cost, prevalence and health outcomes between urban and rural places.
- The indirect cost of cataract-related blindness is incredibly high compared with the direct cost of screening, diagnosis and treatment in rural setting.

However, these two are merely statements, they are not explaining anything (i.e., why is it “incredibly high”). These do not explain why a higher frequency of screening in the rural setting could decrease the cost.

Reviewer #2:

Remarks to the Author:

Thank you for the manuscript on an important field. I think there is a need for economic evaluations of AI-related technologies, so I find the subject relevant. Please find my comments for the sections of the manuscript below:

1) Abstract: Negative ICERs (ICUR:s) are normally not presented in numbers, as they are difficult to interpret. Further, both ICER and ICUR are used as acronyms, but none of them are spelled out or explained to the reader.

2) Introduction: Only explaining the screening as being based on 'AI, big data, 5G, cloud computing, the IoT, etc' is too vague, in my opinion. If not possible to explain better within this manuscript, please refer to a publication or appendix where this is explained at least on a superficial level.

3) Results & discussion: Just as in the abstract, negative ICERs are being presented in exact numbers with no explanation of the reason for being negative. The conclusions will vary depending on whether negativity stems from costs or QALYs.

On line 116, you refer to the current WTP threshold. As WTP thresholds as such are varying between countries (and a research area as such), I think a definition of which threshold you use and a reference to why this is the correct one to use in this study. Further, in the discussion on line 130, you refer to the latest WTP threshold. Is that the same as mentioned before? Reference is needed!

4)Methods: I miss a discussion on compliance.Compliance is mentioned later on but as far as I understand, its importance for the model/results is not being touched upon.

My main issue with this study is the utility values being used. A utility level of 1 for people in the age of 50 and above (or any age) is not plausible. That would mean that all people without cataract are perfectly healthy. How does this assumption affect the results of the model?

Is there any reference available for the assumption of 98% having a smartphone (line 284) ?

On line306, you say that a negative ICER indicates a dominating strategy. That might be the case here, but is not true in general. A more expensive strategy producing QALY losses would also result in a negative ICER,

Point-by-Point Response to Reviewers

Manuscript ID: NCOMMS-23-32681A

Re: Cost-effectiveness and cost-utility of a digital technology-driven hierarchical healthcare screening pattern in China: A decision-analytic Markov model

Reviewer #1 (Remarks to the Author):

This is an interesting study that evaluated the cost effectiveness of a unique digital hierarchical (DH) screening pattern for cataract in the community. The model involved 100,000 people in the China setting. They concluded that DH screening is cost-effective in urban and rural China, and this provide a solid model for Low- and middle-income countries (LMICs). However, there are several fundamental problems:

1. The DH system was not described with enough details. How would the participants be selected (apart from the age)? Are they participating by volunteering means? If that is the case, would that be any selection bias (e.g. residents that do not think they have any eye problem will not join the programme).

Response: Thank you for your valuable comments.

First, our research is a simulated economic evaluation based on decision-analytic Markov models for the DH screening pattern with reference to previous research¹⁻⁵, rather than a real-world population-based trial. It has been described in the abstract and context:

Abstract: A simulated cohort of 100,000 individuals ... (Line 41-42)

Methods: The model was built on a simulated cohort of 100,000 residents... (Line 217-218)

Second, we tried to set the simulation screening scenario as close to the real situation as possible to decline selection bias. In this simulated population, except for those who cannot cooperate to take pictures of ocular anterior segments using smartphones or be assisted by their family member, all elderly people aged 50 and above living in the community are invited to participate in the screening program. The data mimicked the situation in which residents are educated before participation that regular eye screening is recommended even if there are no obvious ophthalmic symptoms for elderly people by poster, online advertisements, or trained volunteers in the community clinics, which could help recruit as many participants as possible and avoid selection bias. Additionally, the participation rate is considered for those failing to join the regular screening for subjective reasons. Relevant information has been added to our revised manuscript in the Methods, Line 243-258.

What is exactly the rate of detecting mild, moderate, and severe cataract with the DH system. How correlative would that be with the real-world situation.

Response: Thank you very much. In DH smartphone screening, the sensitivity and specificity of detecting referable cataracts (moderate and severe cataracts) from others (normal and mild cataracts) were 88.67% and 89.33%, respectively. In DH/AI community screening, the sensitivity and specificity are 94.8% and 97%, respectively, shown in Supplementary Table 3. The data were all

derived from a Chinese population-based real-world external validation.⁶ Furthermore, to consider utilities and treatment costs related to different severities of cataracts, we calculated the number of each state using the prevalence and proportions of cataracts from Chinese large-scale epidemiological and meta-analysis data, displayed in Supplementary Table 2.⁷⁻⁹

2. The authors mentioned that the referral adherence of home-based self-screening could be around 62%. However, who about those who did not adhere to the referral? Would there be a delay in treatment that could also be cost if they don't do so?

Response: Thank you for your questions. There are three possible endings for those who fail to adhere to the referral. First, they miss referrals and treatment, leading to decreased utility and increased costs for the treatment of more severe cataracts or even indirect costs for blindness. Second, they can be opportunistically diagnosed and treated as presenting at the hospital for another concern, without routine ophthalmic screening. Third, they can be diagnosed and adhere to the referral and treatment in the next screening cycle. In the original article, we have listed all possibilities and taken the corresponding costs into consideration. Yet, the information scattered throughout the manuscript, making important messages separate. In the revised edition, we added the statements of the abovementioned possibilities in the Methods, Line 301-303 to be more understandable:

“For those who fail to participate in the screening program, or don't adhere to referral or treatment, the possible results can be natural progression of cataracts; otherwise, they can also be diagnosed and treated in opportunistically case finding or next screening cycle.”

And for those who are screened and are told to have no cataract, would they think they no longer have any eye problem and miss out other eye diseases that could lead to utility loss (e.g. an asymptomatic glaucoma patient who have been told by the DH system that don't have cataract, ended up missing out the diagnosis of glaucoma and the chance of treatment at the early stage, ended up having permanent utility lost because of delay treatment). Did the model consider that? And if it did, what are the percentages of missing out those chronic diseases (e.g. glaucoma, ARMD).

Response: Thank you for your constructive comments.

The priority propose of this study is to propose a novel DH screening pattern, which has the potential to be further applied to screening, monitoring, and managing suitable diseases in a home-based setting. In the preliminary exploration, we selected cataract as the first screening target to illustrate the cost-effectiveness of the system, for the following reasons. First, cataracts are the leading cause of vision impairment and blindness globally. Second, cataracts are irreversible blindness and patients' visual acuity can be improved effectively after proper treatment. Therefore, regular screening and early treatment are necessary. Third, cataracts are visible and easy to take photographs using smartphones without extra attachment. Taking ARMD as an example, a smartphone-based retinal camera apparatus is needed which is expensive and not yet in widespread use. Therefore, we take cataract as the first example to conduct economic evaluations of the DH system and assume that there are no other ocular diseases combined in the simulated population of Markov models.

Although only cataract is considered in this simulated analysis, we would replenish the workflow to address this ethical issue, meanwhile summarize the percentages of missing out those chronic diseases (e.g. glaucoma, ARMD) in preparation for the next study involving multiple diseases screening. For instance, once the participants are presented with nonreferral mild cataract but accompanied by unmatched poor vision (suspicious of other ocular diseases), the trained medical staff in the community-based clinic will give a reminder of further referral and comprehensive examinations, thus avoid missing out other chronic eye diseases.

Nevertheless, it is one of the limitations of our study that only one eye disease is involved, which has been mentioned in the Discussion section in Line 195-197: “Comprehensive ocular examinations including the anterior segment and retinal fundus are more practical and efficient in real-world applications. Multiple eye disease screening will be considered in our future research.” Next, we plan to involve multiple ocular diseases including glaucoma, ARMD, and diabetic retinopathy to further confirm the cost-effectiveness of comprehensive blindness-causing eye disease DH screening scenarios.

Again, we appreciate your insightful suggestions.

3. There are too many supplementary materials that could confuse the readers, and this is probably not the best way to get away from the “table + figure” restriction set by the journal. For instance, the cost eTable 5-8 could have been combined into one table instead of three. Information of eFigure 4 is probably redundant because it is commonly known among cost-effectiveness analysis and could be described in the manuscript.

Response: We appreciate your kind comments. As suggested, eFigure 4 has been deleted, and relevant information has been added to the Methods section in Line 316-319. The eTables 6-7 have been combined into one table (Supplementary Table 6) in the revised Supplementary Information.

However, we think it may be reasonable to keep eTable 5 (Supplementary Table 5, in the revised edition) independent since it focuses on different cost comparisons from Supplementary Table 6. Supplementary Table 5 shows the screening costs of DH/AI screening and telemedicine screening. Supplementary Table 6 lists the differences in the costs of examinations, treatment and follow-up of normal people and patients with different stages of cataracts. Therefore, we believe it may be more understandable to list them in separate parts.

4. The manuscript has a lot of information but the flow of the whole layout and discussion could make it difficult to follow.

Response: Thank you for bringing these important concerns to our attention. To improve the flow of the whole layout of the manuscript, we added subtitles in the Results section to make it more understandable and adjusted the structure in the Discussion section. If there are any other specific suggestions to help illustrate the whole workflow of the article, we are happy to make changes for improvements.

5. Though per capita GDP is the standard way of setting the WTP. However, China’s GDP make up is quite different from the European countries and the United State of America. Generally, the

Europe and US has a larger proportion of the GDP taken up by the consumption market and lesser proportion occupied by investment and infrastructure. The consumption portion is more related to the actual money that the people have and willing to spend. However, this is the other way round in in China; a larger proportion of the GDP is taken up by the investment and infrastructure and lesser proportion is occupied by consumption. Hence, the “money in the people’s pocket” is not reflected by the GDP in the same way as the other countries. The standard per capital GDP may not be the best way to derive WTP in the China setting. So much as that the author could derive the WTP of China as per the customary method, the authors should point this out in the discussion.

Response: We appreciate your insights to this important issue. Per capita GDP is widely used as the standard way of setting WTP in cost-effectiveness analysis worldwide,^{1-5,10-15} which is a recommended threshold from the Commission on Macroeconomics and Health of the World Health Organization (WHO).¹⁶ As you have mentioned, the proportion of the consumption market in China’s GDP is less than that in the United States and European countries. We point this issue out in the Discussion section as suggested in Line 203-207:

“Third, although per capita GDP is the standard way of setting WTP thresholds in cost-effectiveness analysis, the conditions in China are different. China has a smaller proportion of the consumption market in GDP than the United States. Thus, the “money in the people’s pocket” is not reflected by the GDP in the same way as the other countries and a better way to derive WTP in the Chinese setting is needed.”

Other comments

Introduction

(lines 86-87): Although the statement that “mild and severe cataracts are visible through external appearance capture by mobile terminals, making it possible for home-setting screening on a large scale.” is supported by previous publication. However, small pupil and poor quality of photo taking (common for untrained hands) are common. Did the authors take those into consideration and the cost that could lead to?

Response: Thank you for your comments. For the smartphone-based AI screening model, photographs of small pupils are included in the model development and evaluation.⁶ Therefore, the performance data fit into real-world scenarios with nonmydriatic photographs contained. Additionally, there are video and text instructions on the mini app about how to take pictures of ocular anterior segments with useable quality for patients or their family members. These spendings have been added to the AI platform costs.

Methods:

(line 209) A Markov model simulated from 50 years for 30 years – Does that fit into the life expectancy in China? Furthermore, If the 100,000 residents start from 50 years and begin to do the screening, why would a cataract only catch at 80 years old? (Although this is based on previous eye disease screening studies).

Response: Thank you very much.

For the first question, China's average life expectancy is 78.2 years in 2021.¹⁷ Age-related mortality was considered in our analysis.¹⁸ Each participant in the model could progress to death at any health state in the model with reference to age-related mortality from the China Population Census Yearbook 2020 from the National Bureau of Statistics.¹⁸ A Markov model simulated from 50 years for 30 years is a common target population based on previous economic evaluations of eye screening in Chinese elderly people.^{2,3,5}

For the second question, there are two possible ways for a participant to be diagnosed with cataracts at 80 years old. First, he or she was free from cataracts before 80 years old and had late-onset cataracts at 80, which were diagnosed in the screening program. Second, he or she suffered from cataracts before 80 years old but failed to adhere to screening, referral, or treatment, or to be treated through opportunistic findings, which resulted in being untreated until he or she was finally diagnosed and treated in the screening program at 80 years old.

How was the severity of cataract assessed exactly? Seems like not merely based on patients BCVA (line 215 to 216)? Because it was also mentioned in the telescreening part that an ophthalmologist assessed the severity of the cataract for feed back to the primary care settings? (lines 233-234) – authors need to clarify about that point.

Response: Thank you for your valuable advice. Based on the clinical practice guidelines, the severity of cataracts is assessed by slit-lamp photographs using LOCS grading standards.¹⁹ Increased cataract severity is strongly associated with a decrease in visual acuity.²⁰ Therefore, cataract patients' BCVA is one of the common classification methods in clinical trials.^{5,8,21} We derived data of mild, moderate and severe cataracts based on patients' best corrected visual acuity (BCVA) >0.3 , $0.1-0.3$, and <0.1 , respectively from published research.^{8,17} In the telescreening group, ophthalmologists assess the severity of the cataract based on patients' slit-lamp photographs with reference to visual acuity and give the final diagnosis back to the primary care settings. Relevant explanations have been added in the Methods, Line 223-228:

“Based on the clinical practice guidelines, the severity of cataracts is assessed by slit-lamp photographs using LOCS grading standards. Increased cataract severity is strongly associated with a decrease in visual acuity. Therefore, cataract patients' BCVA is one of the common classification methods in clinical trials. We derived data of mild, moderate, and severe cataracts based on patients' best corrected visual acuity (BCVA) >0.3 , $0.1-0.3$, and <0.1 respectively from published research.”

How about those who were invited but did not participate in DH screening? Percentage of that? If patient deny for DH screening, would that be the cost as well?

Response: The participation rate of DH screening is 98%, as listed in Supplementary Table 3. If the patient denies DH screening, there would be no direct screening costs. However, if the cataracts progress into a more severe stage because of delays in screening and treatment, there may be more following treatment costs or indirect costs due to blindness.

Did the author consider the cost of miss out other eye disease (and the utility loss of that) because

the participants feel comfortable with the DH system telling them that do not have cataract and did not seek ophthalmologist/optometrist help even if the visual acuity worse?

Response: Thank you for pointing out this important issue.

The purpose of this study was to explore the economic evaluation of a novel DH screening pattern; therefore, a typical eye disease – cataract, was selected as an example to evaluate the cost-effectiveness of the system. Therefore, we assume that there are no other ocular diseases combined. We have discussed the limitation in the Discussion section in Line 197-203: “Second, comprehensive ocular examinations including the anterior segment and retinal fundus are more practical and efficient in real-world applications. Multiple eye disease screening will be considered in our future research. Although only cataract is considered in the simulated analysis, we would replenish the workflow to address this ethical issue, meanwhile summarize the percentages of missing out those chronic diseases (e.g., glaucoma, age-related macular degeneration) in preparation for the next study involving multiple diseases screening. For instance, once the participants are presented with nonreferral mild cataract but accompanied by unmatched poor vision (suspicious of other ocular diseases), the trained staff in the community-based clinic will give a reminder of further referral and comprehensive examinations, thus avoid missing out other chronic eye diseases.”

However, in our future study, we are working on including more blindness-causing eye diseases, such as glaucoma, DR, AMD, and myopic maculopathy, to further prove the cost-effectiveness of the multiple eye disease DH screening system.

“We assumed that patients’ visual acuity and utility remained stable after surgery” (line 222) How about PCO after surgery? How about other potential complications. How about complications of cataract and the related management.

Response: Thank you for your comments. The main purpose of this study is to illustrate the cost-effectiveness of a novel DH screening pattern. Therefore, we simplified the treatment process as previously described^{2,3,5}, without taking postoperative complications into account. This is one of the limitations that we mentioned in the Discussion section, Line 190-195:

“First, the process of disease treatment was simplified in illustrating the workflow of the DH screening strategies. Cataracts were selected for analysis as they are recognized as the leading cause of vision loss and blindness worldwide. Additionally, it was presumed that patients with bilateral cataracts would demonstrate similar levels of severity and postoperative visual outcomes. The costs and utility might vary if dissymmetric cataracts or postoperative complications are considered.”

Authors should rearrange the flow of describing the study methodology. Information tended to scatter within the section and that could be confusing to the readers.

Given that the cost-effectiveness analysis was based on national cataract AI screening programme implemented, authors should clarify the details and terms related to the programme before describing the cost-effectiveness analysis based on the specific terms.

For instance

- POST-1, POST-2 and POST-3 groups were mentioned under Model Overview (line 219 to 210)

with referral to a supplementary figure (eFigure 3), while the details were only described scattered throughout the section. That makes it very difficult to follow.

Response: We apologize for the original layout with important information scattered. We have clarified the definitions of these terms in the Methods section, Line 223-233:

“Based on the clinical practice guidelines, the severity of cataracts is assessed by slit-lamp photographs using LOCS grading standards. Increased cataract severity is strongly associated with a decrease in visual acuity. Therefore, cataract patients’ BCVA is one of the common classification methods in clinical trials. We derived data of mild, moderate and severe cataracts based on patients’ best corrected visual acuity (BCVA) >0.3 , $0.1-0.3$, and <0.1 respectively from published research...Accordingly, we defined three postoperative groups based on patients’ BCVA after surgery, namely the POST-1 group (BCVA >0.3), the POST-2 group (BCVA $0.1-0.3$), and the POST-3 group (BCVA <0.1) for utility analysis.”

- They mentioned about “AI performances” (line 254) at the later part of the session. That makes it confusing again because it seems unclear whether they are referring to the AI in the “AI Screening” or the AI in the “DH screening”.

Response: Thank you for pointing out this issue. To make this clear, we have replaced the expression “AI performances” with “The model performances of DH smartphone-based screening and community-based AI/DH screening” in the Methods section, Line 266:

“The model performances of DH smartphone-based screening and community-based AI/DH screening were derived from an ongoing national cataract AI screening investigation launched by the ZOC in 2018 to promote collaborative efficiency and medical resource coverage.”

The authors did not mention about inflation in the calculation.

Response: Thank you for your comment. Per National Institute for Health and Care Excellence recommendations, health costs and utility weights were discounted at an annual rate of 3.5%.⁵ The discount rate of 3.5% representing the inflation is considered in the calculation, which is widely used in cost-effectiveness analysis.^{1,2,5}

Results:

The results part is related to the assumption of the model that the authors made. Therefore, the problem in the results is corresponding to that of the methodology.

Response: Thank you very much. After the abovementioned adjustments of the Methods section, we made corresponding changes and added subtitles to make them more understandable in the Results section.

Discussion:

The discussion tried to be too inclusive about various aspect of AI screening and divert from what really need to discuss about the results of the study. The authors spent a lot of effort discussing about the advantage of home screening and AI of other eye diseases (without much disadvantage),

spending a paragraph to emphasize on the same point, but the discussion of the important parts was a bit anaemic. For instance, (line 177-179) they mentioned that increasing frequency of screening in the rural settings gained more benefits while the cost decrease. The author explained that the disparity (compared with the urban setting) was possible due to

- the difference in cost, prevalence and health outcomes between urban and rural places.
- The indirect cost of cataract-related blindness is incredibly high compared with the direct cost of screening, diagnosis and treatment in rural setting.

However, these two are merely statements, they are not explaining anything (i.e., why is it “incredibly high”). These do not explain why a higher frequency of screening in the rural setting could decrease the cost.

Response: We appreciate your valuable suggestions. To improve the logic of discussion, we added more precise and detailed explanations for the disparity of different screening intervals and related costs in urban and rural settings in the Discussion section, Line 177-182:

“The main reason is that more frequent DH screening could save more indirect costs due to blindness in rural areas. First, the prevalence of cataracts and the proportions of severe cataracts are higher in rural areas. Second, due to relatively limited medical resources, the opportunistic case finding rate is lower than that in urban settings. Rural patients would benefit from more frequent screening programs for being spotted and treated; hence, total societal spending was saved by avoiding utility loss and blindness-related indirect costs.”

Reviewer #2 (Remarks to the Author):

Thank you for the manuscript on an important field. I think there is a need for economic evaluations of AI-related technologies, so I find the subject relevant.

Response: We thank the reviewer for the positive feedback on our work.

Please find my comments for the sections of the manuscript below:

1) Abstract: Negative ICERs (ICURs) are normally not presented in numbers, as they are difficult to interpret. Further, both ICER and ICUR are used as acronyms, but none of them are spelled out or explained to the reader.

Response: We appreciate your valuable comments. As suggested, all numbers indicating negative ICER/ICUR were removed in the revised edition. In addition, we added the full spelling and explanations of ICER/ICUR in the Abstract in Line 42-43.

“The primary outcomes were incremental cost-effectiveness ratio and incremental cost-utility ratio.”

2) Introduction: Only explaining the screening as being based on ‘AI, big data, 5G, cloud computing, the IoT, etc’ is too vague, in my opinion. If not possible to explain better within this manuscript, please refer to a publication or appendix where this is explained at least on a superficial level.

Response: We thank the reviewer for bringing out the important concern. To better illustrate the workflow of DH screening pattern, we elaborated on the definitions and added relevant references in Introduction, Line 61-64 and Reference 12-14:

“To further promote the equity and accessibility of health care, a digital technology-driven hierarchical (DH) screening pattern has been proposed. It consists of AI screening and diagnosis models based on multiple examination devices, with cloud computing and IoT facilitating telecommunications.”

3) Results & discussion: Just as in the abstract, negative ICERs are being presented in exact numbers with no explanation of the reason for being negative. The conclusions will vary depending on whether negativity stems from costs or QALYs.

Response: Thanks for your comments. All negative ICER/ICUR in this article came from fewer costs with gained benefits, which is called “dominating” generally. To avoid the unnecessary misunderstanding, we adjusted the expressions of “dominating” definitions in Methods part in Line 332-322:

“Notably, if the ICUR or ICER was negative with fewer costs spent and more benefits gained, the strategy is defined as *dominating*.”

On line 116, you refer to the current WTP threshold. As WTP thresholds as such are varying between countries (and a research area as such), I think a definition of which threshold you use

and a reference to why this is the correct one to use in this study. Further, in the discussion on line 130, you refer to the latest WTP threshold. Is that the same as mentioned before? Reference is needed!

Response: Thanks very much. We referred to the recent published cost-effectiveness analysis in China and most used the local per capita GDP in 2019 as WTP thresholds.^{1-5,12-14} The formulas are as follows.^{2,3,5}

The per capita GDP of urban China

$$= \frac{\text{overall per capita national GDP}}{\left(1 + \frac{1}{\text{urban to rural ratio of per capita disposal income}}\right) \times \text{urbanization rate}}$$

The per capita GDP of rural China

$$= \frac{\text{overall per capita national GDP}}{\left(1 + \text{urban to rural ratio of per capita disposal income}\right) \left(1 - \text{urbanization rate}\right)}$$

In our analysis, we used the latest data including 2022 overall per capita national GDP (\$12,741), urbanization rate (0.65), and urban-rural ratio (2.45) to set the current WTP thresholds, instead of using old data in 2019. References are added as suggested in Methods, Line 311-315.^{22,23}

4)Methods: I miss a discussion on compliance. Compliance is mentioned later on but as far as I understand, its importance for the model/results is not being touched upon.

Response: Thanks for your comments. In our one-way sensitivity analysis, compliance doesn't rank top five most significant parameters in the model (Figure 3), therefore its fluctuations are not as important as indirect costs for blindness, screening costs, utility, and prevalence of cataracts.

My main issue with this study is the utility values being used. A utility level of 1 for people in the age of 50 and above (or any age) is not plausible. That would mean that all people without cataract are perfectly healthy. How does this assumption affect the results of the model?

Response: In previous published paper of eye screening cost-effectiveness analysis, the utility of a 50-year-old person free from targeted eye diseases has been set as 1.^{2,3,5} However, it is not plausible to assume an elderly patients without cataracts are perfectly healthy. It is complicated to quantify the exact utility when other ocular and systematic diseases with various stages combined are considered. Therefore, we select utility of 1 for participant free from cataracts in the analysis based on previous research.^{2,3,5} Additionally, the discount rate of 3.5% in utility is included in the data analysis to reflect aging process.⁵

Is there any reference available for the assumption of 98% having a smartphone (line 284) ?

Response: Thanks for your kind reminding.

According to the *Statistical Communiqué of the People's Republic of China on the 2022 National Economic and Social Development* released by National Bureau of Statistics of China, there were 1,862.86 million phone subscribers in China by 2022, of whom 1,683.44 million were mobile phone subscribers. The mobile phone coverage was 119.2 sets per 100 persons.²⁴

Additionally, based on the *Digital China Development Report (2022)* issued by the Cyberspace Administration of China at the 6th Digital China Summit in April, 2023, that 5G network has covered 96%-100% rural and urban China by the end of 2022.²⁵ Therefore, considering the coverage of mobile phones and network, we selected 98% as the participation rate representing the proportions of people could have access to home-based DH screening via mini-app.

In our simulated cohorts, even though some elderly individuals do not own smartphone, they can still be screened at home once their family members own one. Thus, we use the participation rate of 98% to represent the actual mobile phone coverage and take exceptional cases (2%) into consideration. Relevant reference has been added in Methods, Line 294.

On line 306, you say that a negative ICER indicates a dominating strategy. That might be the case here but is not true in general. A more expensive strategy producing QALY losses would also result in a negative ICER.

Response: We totally agree with your statements. Since $ICER = \frac{\text{incremental cost}}{\text{years of blindness avoided}}$, there are two possible ways resulting in a negative ICER. The first one is a cheaper strategy producing more QALY, which is called “dominating” strategy generally. The other is a more expensive strategy producing QALY losses. In our analysis, all results producing negative ICER goes into the first situation, therefore we define them as dominating strategies. Thanks for your kind reminders, we made minor changes to the definitions part in Methods, Line 321-322 to make it more precise:

“Notably, if the ICUR or ICER was negative with fewer costs spent and more benefits gained, the strategy is defined as *dominating*.”

Reference

1. Li, R. Implementing a digital comprehensive myopia prevention and control strategy for children and adolescents in China: a cost-effectiveness analysis. **38**, (2023).
2. Li, R. *et al.* Cost-effectiveness and cost-utility of traditional and telemedicine combined population-based age-related macular degeneration and diabetic retinopathy screening in rural and urban China. *Lancet Reg. Health - West. Pac.* **23**, 100435 (2022).
3. Tang, J. *et al.* Cost-effectiveness and cost-utility of population-based glaucoma screening in China: a decision-analytic Markov model. *Lancet Glob. Health* **7**, e968–e978 (2019).
4. Su, S. *et al.* Cost-effectiveness of universal screening for chronic hepatitis B virus infection in China: an economic evaluation. *Lancet Glob. Health* **10**, e278–e287 (2022).
5. Liu, H. *et al.* Economic evaluation of combined population-based screening for multiple blindness-causing eye diseases in China: a cost-effectiveness analysis. *Lancet Glob. Health* S2214109X2200554X (2023) doi:10.1016/S2214-109X(22)00554-X.
6. Wu, X. *et al.* Universal artificial intelligence platform for collaborative management of cataracts. *Br. J. Ophthalmol.* **103**, 1553–1560 (2019).
7. Huang, S. Prevalence and Causes of Visual Impairment in Chinese Adults in Urban Southern China: The Liwan Eye Study. *Arch. Ophthalmol.* **127**, 1362 (2009).
8. Tan, X. *et al.* Impact of cataract screening integrated into establishment of resident health record on surgical output in a rural area of south China. *Ann. Transl. Med.* **8**, 1222–1222 (2020).
9. Du, Y.-F. *et al.* Prevalence of cataract and cataract surgery in urban and rural Chinese populations over 50 years old: a systematic review and Meta-analysis. *Int. J. Ophthalmol.* **15**, 141–149 (2022).
10. Losina, E. *et al.* Cost-Effectiveness of Preventing Loss to Follow-up in HIV Treatment Programs: A Côte d'Ivoire Appraisal. *PLoS Med.* **6**, e1000173 (2009).
11. Uppal, A., Rahman, S., Campbell, J. R., Oxlade, O. & Menzies, D. Economic and modeling evidence for tuberculosis preventive therapy among people living with HIV: A systematic review and meta-analysis. *PLOS Med.* **18**, e1003712 (2021).
12. Zhang, S. Cost-effectiveness of expanded antiviral treatment for chronic hepatitis B virus infection in China: an economic evaluation. *Lancet Reg. Health - West. Pac.* **35**, (2023).
13. Li, R. *et al.* Cost-utility analysis of commonly used anti-glaucoma interventions for mild-to-moderate primary open-angle glaucoma patients in rural and urban China. *BMJ Open* **13**, e073219 (2023).
14. Shen, M. Cost-effectiveness of artificial intelligence-assisted liquid-based cytology testing for cervical cancer screening in China. *Lancet Reg. Health - West. Pac.* **34**, (2023).
15. Portnoy, A. *et al.* The cost and cost-effectiveness of novel tuberculosis vaccines in low- and middle-income countries: A modeling study. *PLOS Med.* **20**, e1004155 (2023).
16. *Macroeconomics and health: investing in health for economic development ; report of the Commission on Macroeconomics and Health.* (World Health Organization, 2001).
17. The State Council of the People's Republic of China. China's average life expectancy rises to 78.2 years. http://english.www.gov.cn/statecouncil/ministries/202207/13/content_WS62cdfd4dc6d02e533532da48.html (2022).
18. National Bureau of Statistics of China. China Population Census Yearbook 2020. in vols 6–4 (2021).

19. Miller, K. M. *et al.* Cataract in the Adult Eye Preferred Practice Pattern. *Ophthalmology* **129**, P1–P126 (2022).
20. Shandiz, J. H. *et al.* Effect of Cataract Type and Severity on Visual Acuity and Contrast Sensitivity. *J. OPHTHALMIC Vis. Res.* **6**,
21. Huang, W. *et al.* Five-year incidence and postoperative visual outcome of cataract surgery in urban southern China: the Liwan Eye Study. *Invest. Ophthalmol. Vis. Sci.* **53**, 7936–7942 (2012).
22. National Bureau of Statistics of China. *The construction of new urbanization has been solidly promoted, and the quality of urban development has been steadily improved -- the twelfth report of the series of achievements in economic and social development since the 18th National Congress of the Communist Party of China.*
http://www.stats.gov.cn/xxgk/jd/sjjd2020/202209/t20220929_1888803.html (2022).
23. National Bureau of Statistics of China. *The national economy has withstood the pressure and risen to a new level in 2022.* http://www.stats.gov.cn/tjsj/zxfb/202301/t20230117_1892090.html (2023).
24. National Bureau of Statistics. *Statistical Communiqué of the People's Republic of China on the 2022 National Economic and Social Development.*
http://english.www.gov.cn/archive/statistics/202303/01/content_WS63feeda7c6d0a757729e76e8.html (2023).
25. the Cyberspace Administration of China. *Digital China Development Report (2022).*
http://www.cac.gov.cn/2023-05/22/c_1686402318492248.htm (2023).

Reviewers' Comments:

Reviewer #1:

Remarks to the Author:

No further comments

Reviewer #2:

Remarks to the Author:

Dear authors,

Thank you for the clarifications on the comments in the first revision round. From my perspective, two issues remain but apart from that, I am pleased with the revision.

- When it comes to utility, discounting QALYs does not reflect aging, it just reflects the time preference included in economic analysis. You would need to discount even if you had age-adjusted utility weights. It might be that the utility loss from cataract is valid, meaning that results are reliable anyway, but I think you need to elaborate on that as an average utility level of 1 is not in line with quality of life research.

- It is not clear that you actually tested for compliance in the one-way- sensitivity analysis and that it did not end up top 5. The reader might as well think that you did not include that factor at all in the sensitivity analysis. I think it would be good to comment on that.

Point-by-Point Response to Reviewers

Manuscript ID: NCOMMS-23-32681B

Re: Cost-effectiveness and cost-utility of a digital technology-driven hierarchical healthcare screening pattern in China: A decision-analytic Markov model

Reviewer #1 (Remarks to the Author):

No further comments

Response: We appreciate the reviewer for bringing out important concerns to our attention in last revisions, which help our article to be well-organized. Thank you.

Reviewer #2 (Remarks to the Author):

Dear authors,

Thank you for the clarifications on the comments in the first revision round. From my perspective, two issues remain but apart from that, I am pleased with the revision.

Response: Thank you for the positive feedback for our revisions.

- When it comes to utility, discounting QALYs does not reflect aging, it just reflects the time preference included in economic analysis. You would need to discount even if you had age-adjusted utility weights. It might be that the utility loss from cataract is valid, meaning that results are reliable anyway, but I think you need to elaborate on that as an average utility level of 1 is not in line with quality-of-life research.

Response: We appreciate your kind suggestions. We agree that an individual aged 50 or above without cataracts is presumably to have a utility less than 1 for aging process and systematic disorders. We referred previous research of economic evaluation about eye screening in China and found that most still used the utility of 1 for seniors without eye diseases.¹⁻³ Therefore, we added relevant explanations on Methods part, Line 366-369, to point out this issue:

“The utility of healthy individuals without cataracts was set as 1. However, it should be noted that the utility for an individual aged 50 and above is normally lower than 1 for aging process and systematic disorders. But given that only cataract is considered in the analysis, we set the utility of 1 for healthy seniors without cataracts as previous research did.¹⁻³”

and in limitations of Discussion part, Line 266-268:

“Additionally, utility is set as 1 for healthy individuals without cataracts as previous research did.¹⁻³ In future analysis, proper utility should be selected considering aging process and systematic disorders of the senior population.”

Although our results stay reliable with the assumption, we will search for quality-of-life research and found reasonable utility for the target population in future study.

- It is not clear that you actually tested for compliance in the one-way- sensitivity analysis and that it did not end up top 5. The reader might as well think that you did not include that factor at all in the sensitivity analysis. I think it would be good to comment on that.

Response: Thank you for your constructive comments. In the one-way sensitivity analysis, we have tested all the parameters with values and fluctuations listed in the supplementary files, including prevalence, compliance, costs, sensitivity, specificity, utility, etc. Tornado diagrams show the top five parameters that had the greatest influence on the results. In our original manuscript, we failed to point out that we have included all parameters into the analysis which might confuse the readers. We appreciate your kind advice, and added relevant information in Methods part, Line 396-398, to clarify this point:

“All parameters in the model construction listed in the supplementary file are tested for one-way sensitivity analysis to determine which have the greatest influence on the results, including prevalence, utility, compliance, costs, model performances, etc.”

and in Methods part, Line 401-402:

“Tornado diagrams show the top five parameters that had the greatest influence on the ICURs.”

Reference:

1. Tang, J. *et al.* Cost-effectiveness and cost-utility of population-based glaucoma screening in China: a decision-analytic Markov model. *Lancet Glob. Health* **7**, e968–e978 (2019).
2. Li, R. *et al.* Cost-effectiveness and cost-utility of traditional and telemedicine combined population-based age-related macular degeneration and diabetic retinopathy screening in rural and urban China. *Lancet Reg. Health - West. Pac.* **23**, 100435 (2022).
3. Liu, H. *et al.* Economic evaluation of combined population-based screening for multiple blindness-causing eye diseases in China: a cost-effectiveness analysis. *Lancet Glob. Health* S2214109X2200554X (2023) doi:10.1016/S2214-109X(22)00554-X.